# Hydraulic Planning in Insular Urban Territories: The Case of Madeira Island—São Vicente

**Sérgio Lousada** [1,2,3,4,5,*], **Leonardo Gonçalves** [1] **and Alper Atmaca** [6]

1 Department of Civil Engineering and Geology (DECG), Faculty of Exact Sciences and Engineering (FCEE), University of Madeira (UMa), 9000-082 Funchal, Portugal; leonardobazilio13@gmail.com
2 CITUR—Madeira—Research Centre for Tourism Development and Innovation, 9000-082 Funchal, Portugal
3 VALORIZA—Research Centre for Endogenous Resource Valorization, Polytechnic Institute of Portalegre (IPP), 7300-110 Portalegre, Portugal
4 Environmental Resources Analysis Research Group (ARAM), University of Extremadura, 06071 Badajoz, Spain
5 Institute of Research on Territorial Governance and Inter-Organizational Cooperation, 41-300 Dąbrowa Górnicza, Poland
6 Faculty of Engineering, Dumlupinar University, Kütahya 43820, Turkey; alperatmacaa@gmail.com
* Correspondence: slousada@staff.uma.pt; Tel.: +351-963-611-712

**Abstract:** This study aims to examine the flood propensity of the main watercourse of São Vicente drainage basin and, if relevant, to propose two methodologies to alleviate the impacts, i.e., detention basin sizing and riverbed roughness coefficient adjustment. Geomorphological data were obtained from the watershed characterization process and used through the SIG ArcGIS software for the flood propensity assessment and then for the calculation of the expected peak flow rate for a return period of 100 years through the Gumbel Distribution. Subsequently, the drainage capacity of the river mouth was verified using the Manning-Strickler equation, in order to establish whether the river mouth of the watershed has the capacity to drain the entire volume of rainwater in a severe flood event. In summary, it was possible to conclude that São Vicente's watershed river mouth is not able to completely drain the rain flow for the established return period. Thus, its drainage capacity was guaranteed by modifying the walls and streambed roughness coefficient and by sizing the detention basin using the Dutch and the Simplified Triangular Hydrograph methods.

**Keywords:** floods; hydraulics; hydrology; roughness coefficient; territorial management; urban planning

## 1. Introduction

Global warming with its increasing variability leads to an increased risk of both floods and drought [1]. While increases in temperatures are in question for all seasons of the year, precipitation may decrease in one season while increasing in the other season. According to some strong findings, this variability in precipitation will increase even more in the future [2]. This, in turn, will lead to a discontinuity of precipitation throughout the year and, consequently, to an increase in sudden and torrential rains. The most natural consequence of this situation is that floods occur in many basins. Floods, depending on the size of the flow in the surrounding area, affect settlements and agriculture by damaging their areas, lower and upper structures, facilities, and living things, and they interrupt human life and socio-economic activities. Sociological effects on humans from floods, psychological disorders, and the like, are also seriously affected.

The source of water, in addition to determining the amount of water falling on the surface in terms of vegetation characteristics, influences the amount of water on the floor, underground infiltration of ground in terms of soil properties, plant and underground water leaking from the amount of the residual flow to the understanding of the causes of floods and geomorphology relationship is extremely important. There are also incorrect

land uses and engineering structures on stream beds (urbanization, levees, embankments, dams, etc.) that assume important roles in the formation of floods.

As this erroneous and often uncontrolled urban sprawl expands to rural areas, there is obviously the need to put structural and non-structural measures into action to prevent or at least mitigate floods impact [3–5]. A long time ago, the guideline would be to redirect the stream flow, changing the watercourse´s spatial disposition and subsequently its river mouth [5]. However, even though this principle is very effective in the upstream region, it worsens and increases flood risk downstream, therefore only benefiting half of the watercourse, population, and assets whilst risking the other half. This concept does not solve the geomorphological and hydrological problem of the watershed which are often assembled with anthropic pressure. Consequently, there is the need for highly-impact measures to further mitigate floods impact and not only redirect the problem from one area to another [4–7].

Taking all of this into consideration, this study aims to perform a hydrological analysis of the municipality of São Vicente, estimating its peak flowrate for a recurring period of 100 years, and establish a comparison with its watershed´s river mouth drainage capacity. Based on the premise that the streams river mouth hydraulic features are insufficient to drain the expected peak flow rate, it was designed a detention basin to further control the downstream flowrate and avoiding the need to change the stream cross section. This structural measure was also chosen as it results in considerably reduced urban effects and can be complemented with small changes of the streambed and walls roughness coefficient, thus increasing the drainage capacity of the river mouth without affecting its cross section.

## 2. Materials and Methods

### 2.1. Area of Study

This study focuses on the São Vicente's watershed, being located on the northern side of Madeira Island between the latitudes of 32°47′ N and the longitudes of 17°2′ W [8,9]. This watershed is also integrated in the municipality of São Vicente and ends up supplying its own main watercourse, as illustrated in Figure 1.

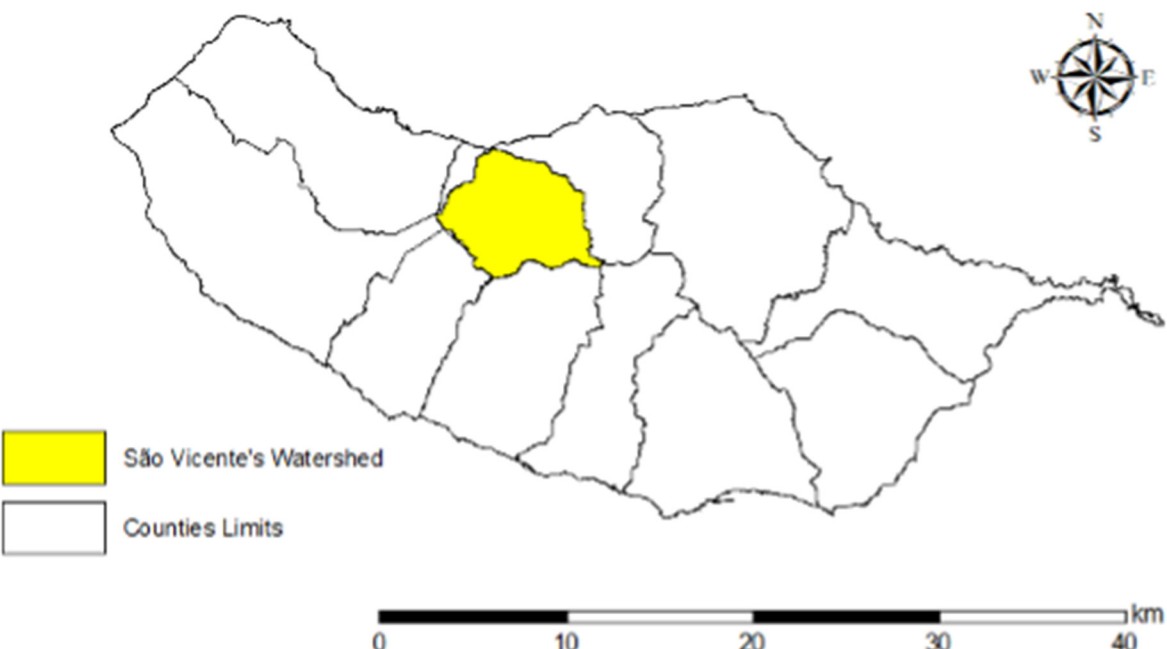

**Figure 1.** The São Vicente watershed. (Source: Authors by ESRI ArcGIS, 2020).

Likewise, the island capital's watersheds—i.e., Funchal, history has recurrently shown that São Vicente's basin suffers with major flooding events, like the ones that took place in 2010 and 2013, resulting in both civil and assets catastrophic losses. São Vicente also suffers

from anthropogenic pressure, much like any other urban municipality, particularly felt by a considerable soil sealing index as a result of an urban sprawl over a semi-rural area [10,11]. Furthermore, its main watershed river mouth is covered in abundant vegetation and a vast track of sediments, that considerably reduce the drainage capacity of this water channel, as shown in Figure 2.

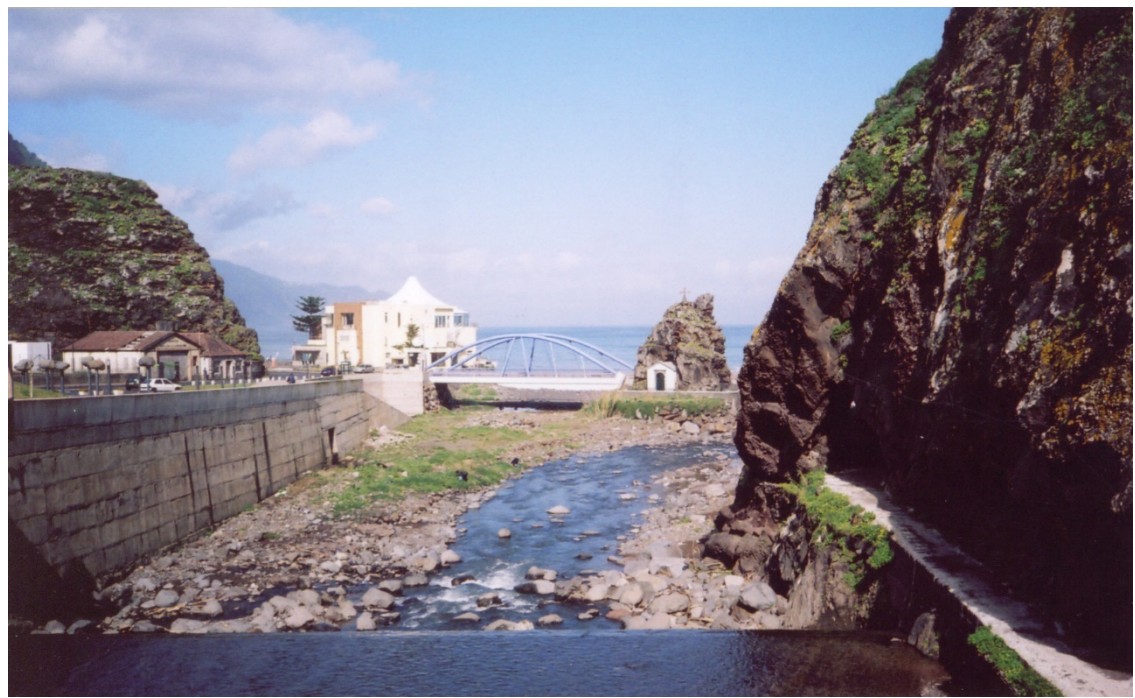

**Figure 2.** State of conservation of São Vicente main watercourse river mouth.

The state of conservation of the stream is virtually the same throughout its length within the urban perimeter, which can be clearly confirmed in situ. Abundant vegetation and sediments can be found along the watercourse, mostly due to the low slope of the streambed that reduces the flowrate's velocity and subsequently the ability to expel sediments through the watershed's river mouth.

### 2.2. Schematic of the Methodology

The methodology adopted can be summarized in 6 phases, as illustrated in Figure 3.

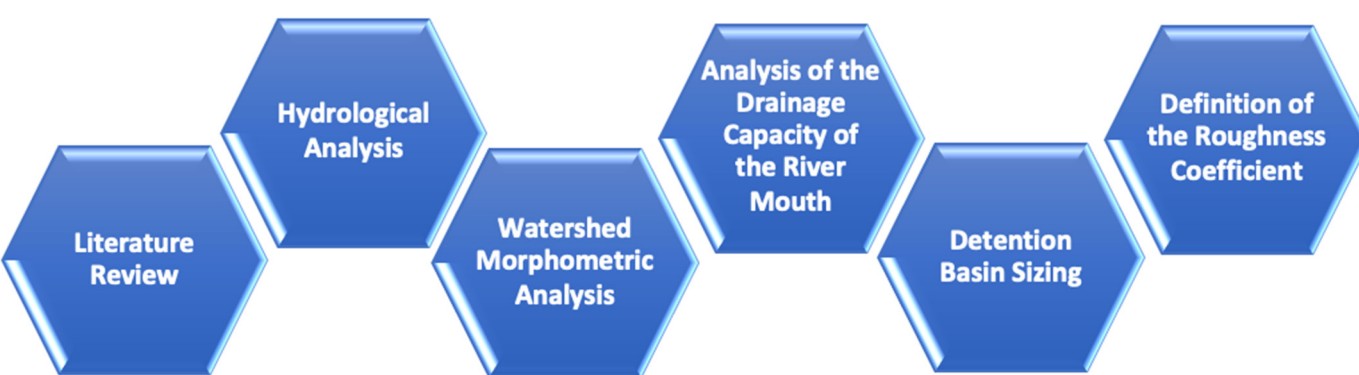

**Figure 3.** Organogram of the adopted methodology.

This case study followed a methodology with six major stages with the first being a deep literature review with the purpose of getting the much needed hydrological and morphometric analysis of São Vicente's watershed. To present a valid contribution for the study

of flood-prone watercourses, different methodologies were also taken into consideration suggested by prestigious authors, thus ensuring the liability of this study's outcome. The following steps are further explained above.

### 2.3. Morphometric Characterization of the Watershed

The key parameters used for the morphometric characterization of a watershed are [4,5,12–17]:

- **Gravelius Index—$K_C$:** This parameter ratio between the perimeter of the watershed with a hypothetical circular watershed with the same area, telling how close to a circular shape the watershed is [13,16]. The Gravelius index can therefore be calculated using Equation (1), being a dimensionless parameter and characterizing the watershed as flood-prone the closer it is to 1 [16].

$$K_C = P/2 \times \sqrt{\pi \times A} \tag{1}$$

where:

P = Perimeter of the watershed, km;
A = Watershed area, km$^2$.

- **Elongation Factor—$K_L$:** The elongation factor determines the ratio between the watershed shape and a rectangle with the same area and can be calculated using Equation (2). It is dimensionless and characterizes the watershed as elongated if the result is higher than 2 [16].

$$K_L = \frac{L_E}{l_E} = \frac{\frac{K_C \times \sqrt{A}}{1.128} \times \left|1 + \sqrt{1 - \left(\frac{1.128}{K_C}\right)^2}\right|}{\frac{K_C \times \sqrt{A}}{1.128} \times \left|1 - \sqrt{1 - \left(\frac{1.128}{K_C}\right)^2}\right|} \tag{2}$$

where:

$L_E$ = Equivalent length, km;
$l_E$ = Equivalent width, km;
$K_C$ = Gravelius Index, dimensionless;
A = Watershed area, km$^2$.

- **Shape Factor—$K_F$:** The Shape Factor describes the ratio between the average width of the watershed with its length. It can be calculated using Equation (3) and is also dimensionless, indicating the basin's elongation degree. The higher is the Shape Factor, the less flood-prone the watershed is and more elongated.

$$K_F = A/L_B^2 \tag{3}$$

where:

A = Watershed area, km$^2$;
$L_B$ = Length of the watershed, km.

The watershed's length can be found measuring the distance between the farthest point to the watershed's river mouth. The watershed's length must not be confused with its main watercourse length, as the last is commonly bigger due to the stream's sinuosity. Using the MDE file provided by the Regional Civil Engineering Laboratory of the Autonomous Region of Madeira (LREC-RAM), it was possible to morphometrically characterize São Vicente's watershed and therefore its main watercourse. The morphometrical data gathered in this study was then processed using different author's methodologies with their very own specific restraints.

A morphometric analysis consists of establishing a hierarchy between the watershed's watercourses—i.e., Strahler or Shreve's hierarchies—according with their order or magnitude [17]. Both types of hierarchy can be conducted following a hydrological analysis of the

DEM file, thus obtaining the "flow accumulation" and the "flow direction" rasters using the "flow order" tool [15]. Nevertheless, the Strahler's hierarchy is deeply associated to a given watershed's bifurcation ratio, with the many degrees of bifurcation being calculated using the Equation (4) [7,12,14–18].

$$R_B = \frac{N_i}{N_{i+1}} \tag{4}$$

where:

$N_i$ = Number of watercourses classified as "i"; dimensionless;
$N_{i+1}$ = Number of watercourses classified as "i + 1", dimensionless.

This dimensionless coefficient is obtained by the ratio of the number of watercourses of a given order by the number of watercourses of the immediately higher order. The average bifurcation value can be calculated based on Equation (5).

$$\overline{R_B} = \sqrt[i-1]{\prod_{i=1}^{i-1} \frac{N_i}{N_{i+1}}} = \sqrt[i-1]{N_1} \tag{5}$$

where:

$N_i$ = Number of watercourses classified as "i"; dimensionless;
$N_{i+1}$ = Number of watercourses classified as "i + 1", dimensionless;
$N_1$ = Number of first-order watercourses.

The bifurcation ratio is also a dimensionless parameter as it merely represents an arithmetic mean of all bifurcation ratios. Moreover, the concentration time of a watershed reveals itself as a key-factor towards the good morphometric characterization of a watershed. It determines the required amount of time needed for all the watershed area contribute to the process of rainfall drainage until it finally crosses the river mouth [12,14,17,18].

Since the equations used to determine the concentration time of a watershed are empirical, each methodology presents different values for the same parameters, and therefore it is advised to use an arithmetic mean of them all, avoiding extremes. In this study, it was used to gather values from the Kirpich (Equation (6)), Témez (Equation (7)), and Giandotti (Equation (8)) methodologies [15].

$$t_C = 57 \times \left( L^3 / (H_{MAX} - H_{MIN}) \right)^{0.385} \tag{6}$$

where:

$t_C$ = Time of concentration, minutes;
L = Length of the main watercourse, km;
$H_{MAX}$ = Maximum height of the main watercourse, m;
$H_{MIN}$ = Minimum height of the main watercourse l, m.

$$t_C = \left( \frac{L}{i^{0.25}} \right)^{0.76} \tag{7}$$

where:

$t_C$ = Time of concentration, hours;
L = Length of the main watercourse, km;
i = Slope of the main watercourse, m/m.

$$t_C = \frac{\left( 4 + \sqrt{A} \right) + (1.5 \times L)}{0.8 \times \sqrt{H_M}} \tag{8}$$

where:

$t_C$ = Time of concentration, hours;
A = Watershed area, km$^2$;
L = Length of the main watercourse, km;
$H_M$ = Average height of the watershed, m.

*2.4. Precipitation Analysis*

The hydrological study is performed based on a probabilistic analysis of extreme high-intensity and short-duration events that took place in São Vicente's watershed throughout history. This data was obtained using National Weather Resources Information System (SNIRH), which also publishes precipitation data recorded automatically in many different stations through the island. Regarding the implemented probabilistic methodology, the Gumbel's Distribution was adopted as it seemed more appropriate to process the obtained data and meet Madeira Island's watersheds projections [19]. Hence, the maximum annual daily precipitation can be calculated using Equation (9).

$$P_{EST} = P_M + S' \times K_T \tag{9}$$

where:

$P_{EST}$ = Estimated annual maximum daily precipitation, mm;
$P_M$ = Average annual precipitation, mm;
$S'$ = Sample standard deviation, mm;
$K_T$ = Frequency Factor, dimensionless.
where:

$$S' = \left( \frac{\sum (X_i - X_M)^2}{n'} \right)^{0.5} \tag{10}$$

where:

$X_i$ = Sample value, mm;
$X_M$ = Sample mean, mm;
$n'$ = Number of samples.

$$K_T = -\frac{6^{0.5}}{\pi} \times \left\{ 0.577216 + \ln \left( \ln \left( \frac{T_R}{T_R - 1} \right) \right) \right\} \tag{11}$$

where:

$T_R$ = Return period, years.

After establishing the daily precipitation for an extreme phenomenon, the precipitation intensity with a particular duration can be obtained using Equation (12).

$$I = \frac{P_{EST} \times k}{t_C} \tag{12}$$

where:

I = Precipitation intensity, mm/h;
$P_{EST}$ = Estimated annual maximum daily precipitation, mm;
$t_C$ = Time of concentration, hours;
k = Time distribution coefficient, dimensionless.
where:

$$k = 0.181 \times \ln(t_C) + 0.4368 \tag{13}$$

where:

$t_C$ = Time of concentration, hours.

The time distribution coefficient is a primary parameter since the annual maximum daily precipitation is only valid for events lasting 24 h. Thus, as the duration of precipitation is equal to the time of concentration of the watershed, using the total amount of daily precipitation in the hydrologic analysis would lead to oversized hydraulic structures [15,20].

*2.5. Drainage Capacity of the River Mouth and Peak Flow Rate*

A river mouth's drainage capacity can be calculated using the Manning-Strickler equation (Equation (14)) and compared to the expected flow for an extreme event with a return period of 100 years. On the other hand, the expected flow for these type of events can also be calculated using the commonly known and used Forti (Equation (16)); Rational (Equation (17)); Giandotti (Equation (18)); and Mockus' (Equation (19)). equations.

$$Q_M = \left(\frac{1}{n}\right) \times A_M \times R^{\frac{2}{3}} \times \sqrt{i} \tag{14}$$

where:

$Q_M$ = Drainage capacity of the river mouth, $m^3/s$.
$A_M$ = Area of the river mouth cross-section, $m^2$;
R = Hydraulic radius, m;
i = Average slope of the river mouth region, m/m;
n = Roughness coefficient of the riverbed and walls, $m^{-1/3}$ s, Table A1.
where:

$$R = \frac{B + 2 \times h}{A_M} \tag{15}$$

where:

B = Width of the river mouth runoff section, m;
h = Height of the river mouth runoff section, m;
$A_M$ = Area of the river mouth cross-section, $m^2$.

The width and height of the stream in the region of the mouth were obtained through previous studies in the region [15], and the first parameter was confirmed via the georeferencing process.

$$Q_{Forti} = A \times \left(b \times \frac{500}{125 + A}\right) + c \tag{16}$$

where:

$Q_{Forti}$ = Peak flow rate by Forti, $m^3/s$;
A = Watershed area, $km^2$;
b = 2.35 for maximum daily precipitation below 200 mm and 3.25 for values above 200 mm;
c = 0.5 for maximum daily precipitation below 200 mm and 1 for values above 200 mm.

$$Q_{Rational} = \frac{C \times I \times A}{3.6} \tag{17}$$

where:

$Q_{Rational}$ = Peak flow rate by the rational methodology, $m^3/s$;
C = Surface runoff coefficient, Table A2;
I = Precipitation intensity, mm/h;
A = Watershed area, $km^2$.

$$Q_{Giandotti} = \frac{\lambda \times A \times P_{MAX}}{t_C} \tag{18}$$

where:

$Q_{Giandotti}$ = Peak flow rate by Giandotti, $m^3/s$;

λ = Reduction coefficient, Table A3;
A = Watershed area, km$^2$;
P$_{MAX}$ = Precipitation height for a duration equal to the concentration time, mm;
t$_C$ = Concentration time, hours.

$$Q_{Mockus} = \frac{2.08 \times A \times P_{EST} \times C}{\sqrt{t_C} + 0.6 \times t_C} \tag{19}$$

where:

Q$_{Mockus}$ = Peak flow rate by Mockus, m$^3$/s;
A = Watershed area, km$^2$;
P$_{EST}$ = Estimated precipitation, cm;
C = Surface runoff coefficient, Table A2;
t$_C$ = Concentration time, hours.

One of the most important design criteria for hydraulic infrastructures is to determine a Fill Rate value lower than 85%, therefore considering a safety margin thus ensuring the safety of the population and their assets [15,21]. Additionally, to control the river mouth's flowrate, it is also required to implement runoff restraining mechanisms, namely spillways.

As mentioned before, the Fill Rate value can be calculated using Equation (20) and if the river mouth's runoff capacity proves to be insufficient to drain the rain flow of the given watershed and respect the proposed safety margin, then it must be designed a flood mitigation mechanism, such as a detention basin.

$$FR = \frac{Q_P}{Q_M} \times 100 \tag{20}$$

where:

FR = Fill Rate, %;
Q$_P$ = Peak flow rate of each methodology, m$^3$/s;
Q$_M$ = Drainage capacity of the river mouth, m$^3$/s.

The Fill Rate parameter refers to the ability of a drainage section to drain a particular flow. Therefore, if the Fill Rate value is greater than 100%, the section is no longer able to drain the total volume of water without overflowing [15].

*2.6. Detention Basin Sizing*

As stated before, if the river mouth cross section is insufficient to drain all the rain flow collected along the watershed, it must be designed some type of spillway to restrain the flowrate, keeping it under the downstream expected limit. In this study, it was adopted a Cipolleti type of spillway as this is a type has features that reduce turbulence regions of contact with the water, thus making it easier to drain all the stream's runoff [5,6,20]. The design of this type of spillways can be performed using the Equation (21).

Knowing the amount of flowrate that must be drained by the stream's river mouth, it is also possible to estimate the volume of water that needs to and will be retained by the detention basin. In this sense, two different methodologies were adopted, namely the Dutch Method (Equation (22)) and the Simplified Triangular Hydrograph (STH; Equation (23)).

$$Q_S = 1.86 \times L_{SD} \times H_D{}^{1.5} \tag{21}$$

where:

Q$_S$ = Flow drained by spillway, m$^3$/s;
L$_{SD}$ = Width of the sill, m$^3$/s;
H$_D$ = Height of the waterline above the sill, m.

$$V_A = (Q_P - Q_S) \times t_c \times 3600 \tag{22}$$

$$V_A = \frac{(Q_P - Q_S) \times (2 \times t_c - 2 \times [Q_S/\{Q_P/t_C\}])}{2} \quad (23)$$

where:

$V_A$ = Storage Volume, m³;

$Q_P$ = Peak flow rate of each methodology, m³/s;

$Q_S$ = Flow drained by the spillway, m³/s;

$t_C$ = Concentration time, hours.

Based on the STH geometric analysis (Figure A1), the Equation (23) was then formulated, considering an event with a duration of at least twice the concentration time of the given watershed. This, took into consideration that the last rain particle to reach the river mouth came from the farthest region and that it also would take place at the last instant of precipitation, indicating that it would need to be equal to the time of concentration to be considered as drained by the river mouth [15].

These methodologies were selected since the Dutch Method does not consider the delay and damping of the precipitation hydrograph, ultimately resulting in the overdesign of the infrastructure [22], as illustrated in Figure 4, where $Q_s$: states for the runoff capacity of the spillway; $t_c$: meaning the concentration time; $t_{MAX}$: being the maximum precipitation duration (standard); $t_d$: being the time delay until the beginning of water accumulation in the detention basin; $H_{a,MAX}$: being the maximum storage capacity; and $i(t_{MAX})$: meaning the precipitation intensity for the maximum duration.

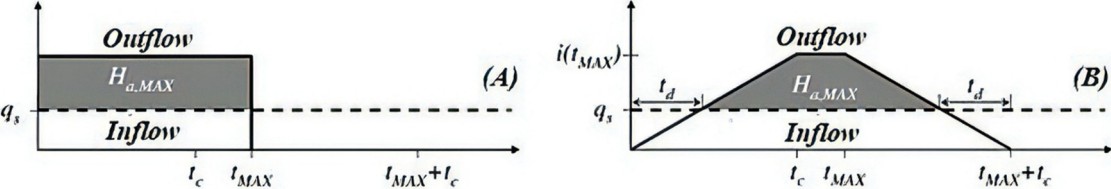

**Figure 4.** (**A**) Dutch method; (**B**) STH method (Source: [22]).

Thus, it was confirmed that, in the Dutch Method, storage begins at the same time with precipitation, which does not correspond to reality as storage will only begin when the flow drained downstream is greater than the spillway's runoff capacity.

*2.7. Modification of the Roughness Coefficient*

Another structural measure that was taken into consideration was the modification of the roughness coefficient of the walls and streambed of the main watercourse, as this would avoid friction between the water and the channel, therefore increasing its drainage capacity. This methodology is based on changing the value of the parameter "n" in the Manning–Strickler equation to improve the flowrate of a given watercourse by considering another type of material or at least its conservation status for its walls and streambed [15].

## 3. Results

The results shown here correspond to the data obtained by applying the formulas already described. Thus, to evaluate the morphometric features of the main watercourse of São Vicente, an individual analysis of each parameter listed in Table 1 was conducted, correlating them with reference values proposed in various bibliographies.

The first parameter refers to the watershed's area, a key-factor for the volume of water that needs to be drained through the river mouth. It can be classified as: Very Large > 20 km²; Large > 10 km²; Medium > 1 km² and Small < 1 km² [23]. This way, as illustrated in Table 1, the considered watershed can be classified as "Very Large", and more flood prone compared to smaller watersheds. Nevertheless, the standard values are arbitrary and may vary according to the analysis conducted throughout the study [23] and with the flood-prone character of the basin.

**Table 1.** Parameters calculated or extracted from ArcGIS.

| Parameter | Unit of Measurement | Value |
|---|---|---|
| Area | km$^2$ | 38.262 |
| Perimeter | km | 37.790 |
| Length of Main Watercourse | km | 10.813 |
| Maximum Height of Main Watercourse | m | 1556.270 |
| Minimum Height of Main Watercourse | m | 0.000 |
| Average Concentration Time | hours | 1.799 |
| Gravelius Coefficient of Compactness | dimensionless | 1.723 |
| Elongation Factor | dimensionless | 7.198 |
| Shape Factor | dimensionless | 0.513 |
| Number of Watercourses | units | 2020.000 |
| Average Bifurcation Ratio | dimensionless | 4.492 |
| Strahler Classification | dimensionless | 6.000 |

As shown in Figure 5, the São Vicente watershed's borders are considerably higher than the central area, indicating a steep slope and subsequently a very fast supply of the main watercourse, thus increasing the volume of water flowing through the stream that will end in its river mouth.

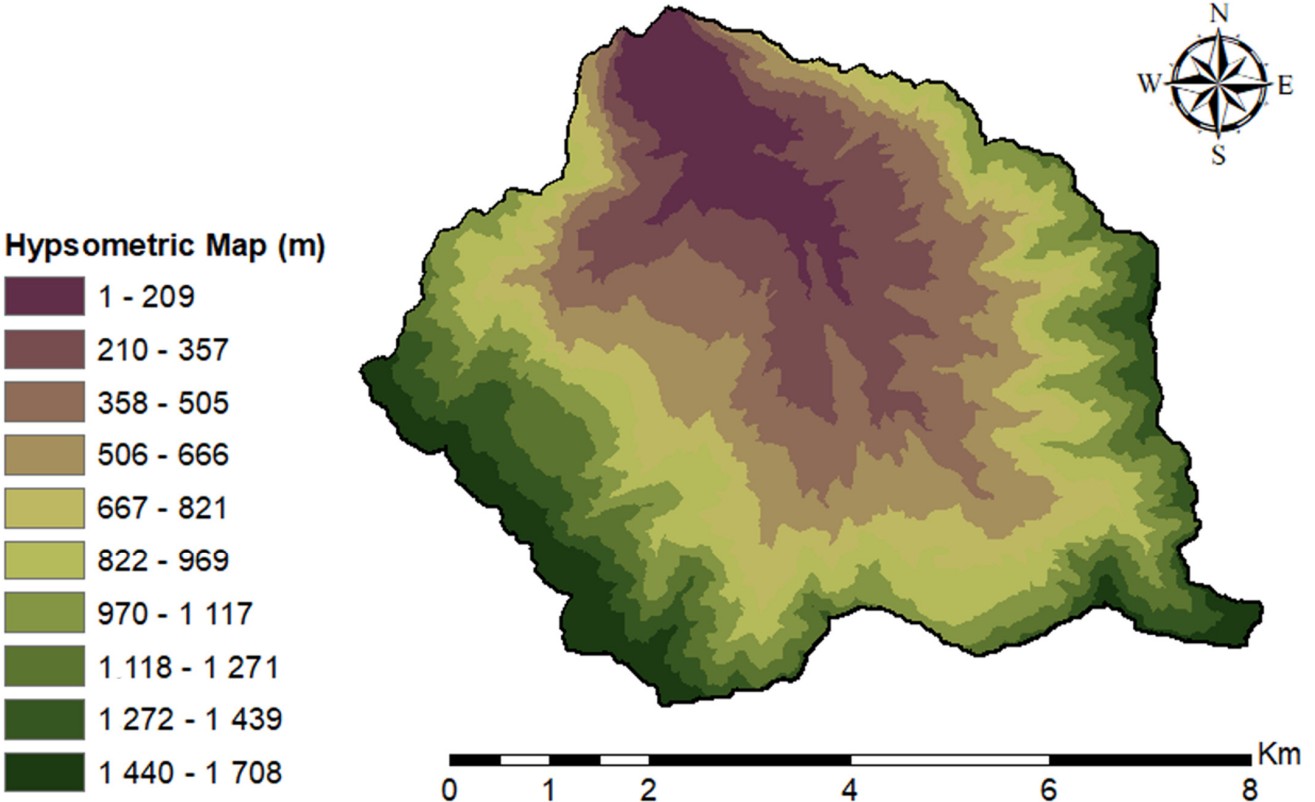

**Figure 5.** Hypsometric map—DEM file (Source: Authors by ESRI ArcGIS, 2020).

Regarding this watershed's singularities, Figure 6 shows a higher number of streams which also suggests that it has a higher drainage capacity—there are many low and medium order tributary streams that supply the main watercourse. In addition, the drainage system is an index that translates to the hydrographic tendency that a watershed has to create new streams. Thus, basins with higher hydric densities tend to present more tributary streams, this happening as a consequence of the ability to generate new streams [15,17].

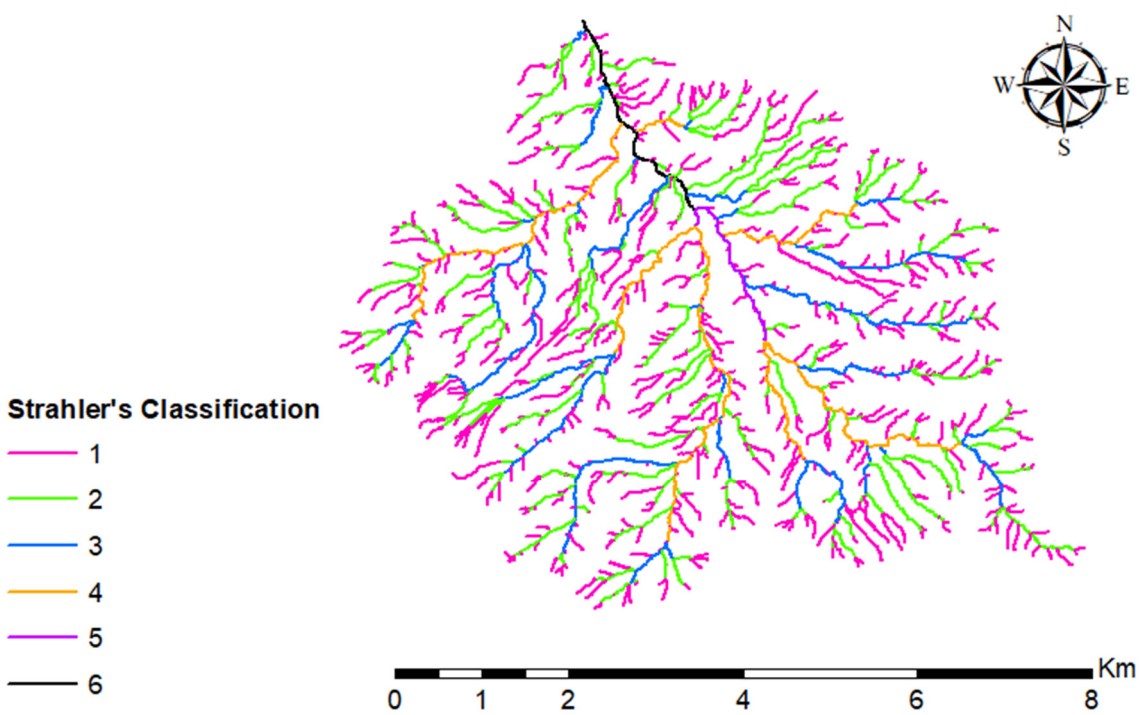

**Figure 6.** Strahler classification (Source: Authors by ESRI ArcGIS, 2020).

Once again, this analysis was only possible thanks to the data available on the National Information System on Water Resources (SNIRH) [24], assessing data samples from a considered period of sixteen years as shown in Table A4 and Figure A2. Therefore, using the Gumbel Distribution's probabilistic process, it was possible to obtain the values shown in Table 2.

**Table 2.** Precipitation parameters.

| Parameter | Symbol | Unit of Measurement | Value |
|---|---|---|---|
| Average Annual Precipitation | $P_M$ | mm | 164.443 |
| Standard Deviation | $S'$ | mm | 64.424 |
| Frequency Factor | $K_T$ | dimensionless | 3.136 |
| Time Distribution Coefficient | k | dimensionless | 0.543 |
| Annual Maximum Daily Precipitation | $P_{EST}$ | mm | 366.521 |
| Precipitation Intensity | I | mm/h | 110.646 |

After obtaining the precipitation intensity index estimated for a return period of 100 years, it was calculated peak flowrates, as presented in Table 3, using the aforementioned methodologies and equations. The surface runoff coefficient particularly used in the rational methodology was 0.500 (Table 4) since the area under study is a peripheral region with commercial buildings. This value translates essentially to the ratio of water that tends to run on the stream's surface, i.e., 50% of the total precipitation.

**Table 3.** Peak flow rate.

| Methodology | Flow (m$^3$/s) |
|---|---|
| Forti | 419.096 |
| Rational | 594.284 |
| Giandotti | 822.796 |
| Mockus | 602.432 |

**Table 4.** Surface drainage coefficient adopted (Source: [25]).

| Urban Areas | | |
|---|---|---|
| **Land Occupation** | | **Surface Drainage Coefficient** |
| Commercial Area | City Center | 0.700–0.950 |
| | Peripheral Areas | **0.500**–0.700 |

The value of the reduction coefficient ($\lambda$) used in the calculation of the flow through Giandotti's methodology is presented in Table 5.

**Table 5.** Adopted Giandotti's reduction coefficient (Source: [26]).

| Area (km$^2$) | $\lambda$ | Equivalent "C" |
|---|---|---|
| <300 | 0.346 | 1.250 |

Regarding the drainage capacity of the considered watershed's river mouth, the Manning–Strickler equation ultimately confirmed the need to further implement a structural flood mitigation measure like the detention basin, where the values obtained are summarized in Table 6. It must also be noted that the walls and the streambed have different roughness coefficients. Consequently, the drainage capacity of the river mouth was calculated through the weighted arithmetic mean of the corresponding coefficients, considering that the stone and mortar walls are in good condition ($n = 0.020$) and the streambed is made of a rocky surface with abundant vegetation in poor condition ($n = 0.040$). Another key-factor that must be taken into consideration is the very low slope of the river mouth, tending to reduce the flowrate velocity and consequently the drainage capacity of the cross section. To simulate and model a critical situation, it was then considered a slope of 0.01 m/m in the reference section.

**Table 6.** Assessment of the need for detention basin implementation.

| Parameter | Unit of Measurement | Value |
|---|---|---|
| Width of the River Mouth | m | 40.000 |
| Height of the River Mouth | m | 3.000 |
| Drainage Capacity of the River Mouth | m$^3$/s | 608.172 |
| Fill Rate—Forti (pre-regularization) | % | 69 |
| Fill Rate—Rational (pre-regularization) | % | 98 |
| Fill Rate—Giandotti (pre-regularization) | % | 135 |
| Fill Rate—Mockus (pre-regularization) | % | 99 |

As presented in Table 6, the Fill Rate is higher than the established limit of 69% for the Rational, Giandotti, and Mockus methods, yet again clearly indicating the need to implement mitigation and flowrate control measures for the river mouth section. Considering this, it was designed a detention basin with the flowrates calculated of the methodologies, affected by the spatial restrains and the anthropogenic pressures of an urban area, namely the already existing infrastructures nearby the watercourse.

Since the design of detention basins depend on the exceeding limits of the flowrate for the watershed's river mouth, a Cipolleti's trapezoid spillway was also designed to restrain and control the flowrate that will end downstream. This type of spillway's features can be found in Table 7.

Afterwards, the detention basins were also designed through the Dutch and the STH Methods, which are merely simplified methodologies that do not take into consideration many key-factors and consequently result in the overdesign of the hydraulic infrastructure. Also, the width and heigh of the detention basin cross section were both fixed with same values of the existing one to reduce this measure's environmental impacts on an urban area.

Thereupon, the only geometric parameter that may vary was its length, being limited to the main watercourse length.

**Table 7.** Application of the Cipolletti spillway.

| Parameter | Unit of Measurement | Value |
| --- | --- | --- |
| Width of the Spillway | m | 38.500 |
| Height of the Spillway Sill | m | 3.000 |
| Spillway Outflow | m$^3$/s | 372.096 |
| Fill Rate—Rational (post-regularization) | % | 61 |
| Fill Rate—Giandotti (post-regularization) | % | 61 |
| Fill Rate—Mockus (post-regularization) | % | 61 |

Using all the methodologies and both methods made possible to present the following length values in Table 8.

**Table 8.** Detention basin sizing.

| Parameter | Unit of Measurement | Value |
| --- | --- | --- |
| Width | m | 40.000 |
| Height | m | 3.000 |
| Length—Dutch Method (Rational) | m | 11,991.486 |
| Length—STH Method (Rational) | m | 4483.318 |
| Length—Dutch Method (Giandotti) | m | 24,324.279 |
| Length—STH Method (Giandotti) | m | 13,324.023 |
| Length—Dutch Method (Mockus) | m | 12,431.234 |
| Length—STH Method (Mockus) | m | 4753.002 |

Finally, changing the roughness coefficient of the streambed and walls was also considered as an alternative flood mitigation structural measure towards preventing its impacts whilst maintaining the same amount and features of the streambed vegetation. This way, Table 9 values were particularly chosen to clearly improve the conservation status of the streambed, thus reducing friction between the drained water and the material covering the watercourse, subsequently increasing its drainage capacity.

**Table 9.** Modification of the roughness coefficient.

| Parameter | Unit of Measurement | Value |
| --- | --- | --- |
| Wall Roughness Coefficient—Modified | m$^{-1/3}$ | 0.012 |
| Riverbed Roughness Coefficient—Modified | m$^{-1/3}$ | 0.030 |
| Drainage Capacity of the River Mouth—Modified | m$^3$/s | 822.371 |
| Fill Rate—Rational (post-modification) | % | 72 |
| Fill Rate—Giandotti (post-modification) | % | 100 |
| Fill Rate—Mockus (post-modification) | % | 73 |

At last, the modified roughness coefficients of the stream walls were considered to have its surface covered in concrete in a good condition status, although the streambed would remain with the same rocky and abundant vegetation features, nevertheless in good condition. The values used for these coefficients are summed in Table 10.

**Table 10.** Adopted roughness coefficient (Source: [26]).

| Channel Typology | Very Good | Good | Regular | Bad |
| --- | --- | --- | --- | --- |
| **Channel with stony and vegetated slope** | 0.025 | **0.030** | 0.035 | 0.040 |
| **Surface with concrete finishing** | 0.011 | **0.012** | 0.013 | 0.015 |

## 4. Discussion

As this study's main goal was to check if it was needed to put into action flood mitigation measures to further prevent major impacts in São Vicente's watershed, the use of a detention basin revealed itself as valid and useful structural measure towards controlling its river mouth's flowrate [27]. At first, the Fill Rate was 98%, 135% and 99%, respectively for Rational, Giandotti, and Mocku's methodologies, which ultimately decreased to only 61% after adopting the detention basin measure. This structural measure's outcome is clear evidence that it may enable the river mouth to work below 85% of its full capacity. Moreover, this proves the accuracy of the Regional Directorate for Territorial Management and Environment (DROTA) prediction, as presented in Table 11.

**Table 11.** Watersheds with high flood risk. (Source: [28]).

| Municipality | Watershed |
|:---:|:---:|
| São Vicente | São Vicente |

This study aimed to cause the least possible impact over the considered watercourse and its surroundings since it is believed that the presence of natural elements in cities present itself and act as a vital condition for the environmental recovery of the urban territory [29]. Additionally, a nature and urban systems symbiosis is typically found as a key-factor or goal to further achieve a territory or city's sustainability [30,31]. Nevertheless, uncontrolled urban sprawl is something that can take place especially in rural areas, thus creating urban voids [32].

As it was not made any change to the stream's cross section, namely its height and width, the only variable parameter was its length. It was based on this concept that the Dutch Method presented an abnormal oversize of the detention basin's length when compared to the watershed's main course's length. Therefore, according to this method it would be needed to change one or both cross section dimensions and so it cannot be considered valid for the aforementioned urban design settings.

The exact same conditions were imposed for the STH method, with it showing a different and this time valid approach since the detention basin's length was shorter than the watershed's main watercourse length.

As for the change of the stream bed and walls roughness coefficient, it was decided to remain with the abundant vegetation and sediments along the watercourse but improving its conservation status by performing a correct maintenance as this would outcome in a cheaper process with less wasting of time and resources. Also, there's no need to perform maintenance on the stream walls frequently, since the mechanical abrasion only happens in alluvial events that tend to result in the drainage of higher volumes of water and large sediments.

Despite being a relatively simple structural measure, the change of the roughness coefficient of this stream, resulted in a significant way, enabling its river mouth to work below the Fill Rate limit, that itself, emphasizing that both methodologies—i.e., the STH method and the changing of the stream's roughness coefficient—can be implemented together, to optimize and reduce the required detention basin's length.

As a final remark, it should be noted that the methodologies were simplified and therefore do not consider local peculiarities. Thus, this may result in oversized hydraulic infrastructures because of conservative considerations and inputs.

## 5. Conclusions

This study revealed how flood-prone São Vicente's watershed is in the event of extreme rainfall occurrence, as it was already predicted by DROTA's own Flood Risk Report. The watershed's drainage capacity is highly decreased by the presence of abundant vegetation and a huge number of sediments throughout the watercourse, resulting in a lower flowrate in an already low-slope stream and river mouth. The insufficient drainage capacity of the

river mouth was verified through 3 of 4 methodologies used in this study namely: Rational, Giandotti, and Mocku's.

Regarding the two methods used during this study, the Dutch Method did not present coherent results as it indicates the need for very long detention basins in relation to the watershed's main watercourse. On the other hand, the Simplified Hydrograph Method presented not only satisfactory results but also to be easier to implement as there is no need for change either the stream's height or width.

Even though changing the watershed's streambed and walls roughness coefficient may seem a relatively simple and unworthy measure to consider, it surely proved to mitigate the flood's impact, fulfilling its main goal by preserving infrastructures and people's assets.

Afterall, this study leaves a clear open-door to others that may complement its contents and methodologies by optimizing its techniques. To improve the often-complex urban hydraulic system and demand, it is also expected that new studies take notice of the need to reduce sediment deposition as it seems to make a huge long-term impact over the watercourse's drainage capacity and ultimately to prevent a major flood impact [33]. On the other hand, mechanical abrasion of this stream's walls and the amount of time that often takes to local public authorities to perform any type of maintenance have been two strong reasons for how degraded the main course tributaries are and subsequently by the lower water quality discharged [34,35] and therefore also need to be studied and improved. Furthermore, the urban growth ratio projected for the municipality of São Vicente and how it may impact soil waterproofing and ultimately surface run-off should be a top concern and studied, complementing this work's outcome.

Generally, this study enhances the methodologies and techniques used in similar case studies as valid and appropriate towards scientific development based on flood scenarios modelling and simulations [36,37].

**Author Contributions:** Conceptualization, S.L., L.G. and A.A.; methodology, S.L.; software, S.L. and L.G.; validation, S.L. and L.G.; formal analysis, S.L. and L.G.; investigation, S.L.; resources, L.G.; data curation, S.L. and L.G.; writing—original draft preparation, S.L.; writing—review and editing, S.L., L.G. and A.A.; visualization, S.L. and L.G.; supervision, S.L. and L.G.; project administration, S.L. All authors have read and agreed to the published version of the manuscript.

**Funding:** This research received no external funding.

**Institutional Review Board Statement:** Not applicable.

**Informed Consent Statement:** Not applicable.

**Data Availability Statement:** The data presented in this study are openly available. Also, it is possible to contact one of the study authors.

**Conflicts of Interest:** The authors declare no conflict of interest.

## Appendix A

**Table A1.** Manning–Strickler roughness coefficients (Source: [26]).

| Type of Channel and Description | Very Good | Good | Regular | Bad |
|---|---|---|---|---|
| Mortared stone masonry | 0.017 | 0.020 | 0.025 | 0.030 |
| Rigged stone masonry | 0.013 | 0.014 | 0.015 | 0.017 |
| Dry stone masonry | 0.025 | 0.033 | 0.033 | 0.035 |
| Brick masonry | 0.012 | 0.013 | 0.015 | 0.017 |
| Smooth metal gutters (semicircular) | 0.011 | 0.012 | 0.013 | 0.016 |
| Open channels in rock (irregular) | 0.035 | 0.040 | 0.045 | - |
| Channels with bottom on land and slope with stones | 0.028 | 0.030 | 0.033 | 0.035 |
| Channels with stony bed and vegetated slope | 0.025 | 0.030 | 0.035 | 0.040 |
| Channels with concrete coating | 0.012 | 0.014 | 0.016 | 0.018 |

**Table A1.** *Cont.*

| Type of Channel and Description | Very Good | Good | Regular | Bad |
|---|---|---|---|---|
| Earth channels (rectilinear and uniform) | 0.017 | 0.020 | 0.023 | 0.025 |
| Dredged canals | 0.025 | 0.028 | 0.030 | 0.033 |
| Clay conduits (drainage) | 0.011 | 0.012 | 0.014 | 0.017 |
| Vitrified clay conduits (sewage) | 0.011 | 0.013 | 0.015 | 0.017 |
| Flattened wooden plank conduits | 0.010 | 0.012 | 0.013 | 0.014 |
| Gabion | 0.022 | 0.030 | 0.035 | - |
| Cement mortar surfaces | 0.011 | 0.012 | 0.013 | 0.015 |
| Smoothed cement surfaces | 0.010 | 0.011 | 0.012 | 0.013 |
| Cast iron coated tube with tar | 0.011 | 0.012 | 0.013 | - |
| Uncoated cast iron pipe | 0.012 | 0.013 | 0.014 | 0.015 |
| Brass or glass tubes | 0.009 | 0.010 | 0.012 | 0.013 |
| Concrete pipes | 0.012 | 0.013 | 0.015 | 0.016 |
| Galvanized iron pipes | 0.013 | 0.014 | 0.015 | 0.017 |
| Rectilinear and uniform clean streams and rivers | 0.025 | 0.028 | 0.030 | 0.033 |
| Streams and rivers cleared rectilinear and uniform with stones and vegetation | 0.030 | 0.033 | 0.035 | 0.040 |
| Streams and rivers cleared rectilinear and uniform with intricacies and wells | 0.035 | 0.040 | 0.045 | 0.050 |
| Spread margins with little vegetation | 0.050 | 0.060 | 0.070 | 0.080 |
| Spread margins with lots of vegetation | 0.075 | 0.100 | 0.125 | 0.150 |

**Table A2.** Surface runoff coefficients (Source: [25]).

| Urban Areas | | |
|---|---|---|
| **Land Occupation** | | **Surface Runoff Coefficient** |
| Green Areas | Lawns in sandy soils | 0.050–0.200 |
| | Lawns on heavy soils | 0.150–0.350 |
| | Parks and cemeteries | 0.100–0.350 |
| | Sports fields | 0.200–0.350 |
| Commercial Areas | City district | 0.700–0.950 |
| | Periphery | 0.500–0.700 |
| Residential Areas | Town-center villas | 0.300–0.500 |
| | Villas on the outskirts | 0.250–0.400 |
| | Apartment buildings | 0.500–0.700 |
| Industrial Areas | Dispersed industry | 0.500–0.800 |
| | Concentrated industry | 0.600–0.900 |
| Railways | | 0.200–0.400 |
| Streets and Roads | Paved | 0.700–0.900 |
| | Concrete | 0.800–0.950 |
| | In brick | 0.700–0.850 |

**Table A3.** Giandotti reduction coefficients (Source: [26]).

| A (km$^2$) | λ | "C" Equivalent |
|---|---|---|
| <300 | 0.346 | 1.250 |
| 300–500 | 0.277 | 1.000 |
| 500–1000 | 0.197 | 0.710 |
| 1000–8000 | 0.100 | 0.360 |
| 8000–20,000 | 0.076 | 0.270 |
| 20,000–70,000 | 0.055 | 0.200 |

**Table A4.** Precipitation historical data (Source: [24]).

| *n* | Year | (mm) |
|---|---|---|
| 1 | 1998/1999 | 170.000 |
| 2 | 1999/2000 | 180.700 |
| 3 | 2000/2001 | 135.000 |
| 4 | 2001/2002 | 190.000 |
| 5 | 2002/2003 | 195.400 |
| 6 | 2003/2004 | 141.000 |
| 7 | 2004/2005 | 103.200 |
| 8 | 2005/2006 | 91.400 |
| 9 | 2006/2007 | 141.400 |
| 10 | 2007/2008 | 104.600 |
| 11 | 2008/2009 | 155.000 |
| 12 | 2009/2010 | 257.800 |
| 13 | 2010/2011 | 148.400 |
| 14 | 2011/2012 | 288.600 |
| 15 | 2012/2013 | 267.400 |
| 16 | 2013/2014 | 61.200 |

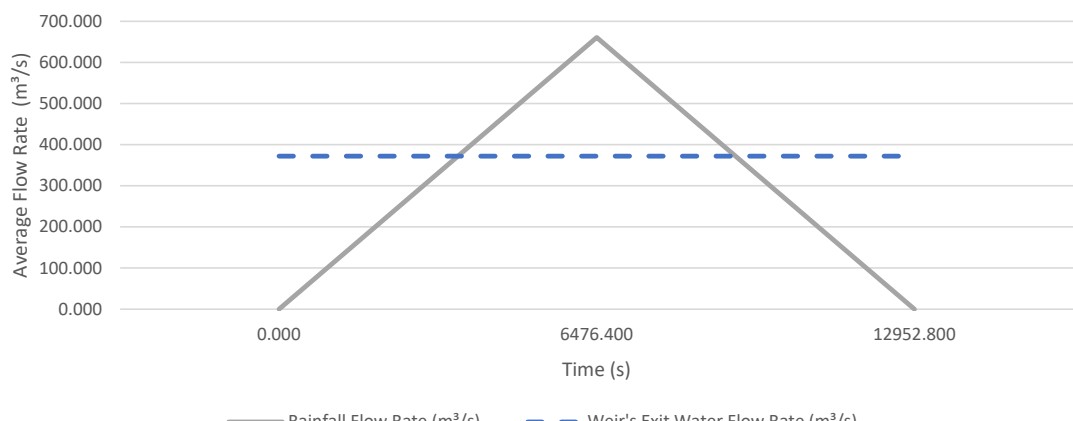

**Figure A1.** Ternary phase diagram.

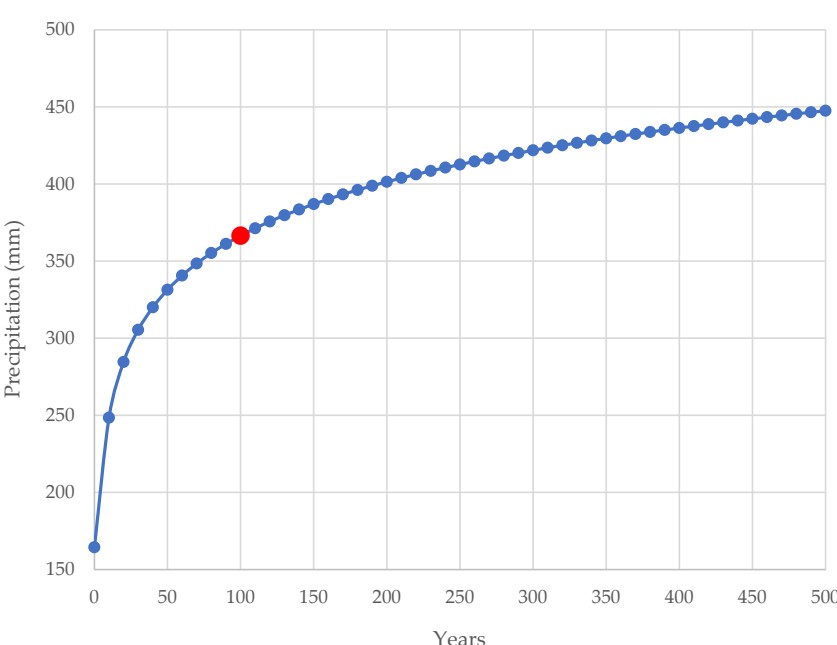

**Figure A2.** Expected rainfall for São Vicente's watershed.

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
