# Peer review of "Hydraulic Planning in Insular Urban Territories: The Case of Madeira Island—São Vicente"

_water, doi:10.3390/w14010112_

Round 1
Reviewer 1 Report
In my opinion, the manuscript is correct as a whole. I have missed a cost-benefit analysis to check whether the investment is socially positive. As it would carry out a long work, I recommend the authors to describe the variables implied and comment that next step to pass the investment would be to make this CBA.
Author Response
Thanks for your comments and effort for providing the report. I agree that in the future an article can be developed to process the financial analysis of these two methodologies, and really see if it will be a positive investment for society.
Reviewer 2 Report
It is well written, but I didn't find anything particularly new from the scientific point of view. The equation used are very simple and can just give a first approximate idea about the storage volume needed to cut the peak discharge. To simulate the real flow it would be better to take into account a real upstream hydrograph and fully implement the continuity equation for the unsteady flow over the weir and for the discharge diverted to the storage capacity.
Author Response
Thanks for your comments and effort for providing the report. The study idea is simple but effective. Because the treatment of precipitation data (daily series of several years) allows to measure the flood peak flow for a return period of 100 years (ideal for hydraulic works). After characterizing the watershed, the processing is realistic, of course within the maximum security for society. One of the variables that can modify the final results is solid transport, however the local authorities regularly process the cleaning of solid debris from the water lines, avoiding the problems that have already occurred in the recent past.
Reviewer 3 Report
The manuscript examined the flood propensity of the main watercourse of São Vicente drainage basin and proposed two methodologies to alleviate the impacts. This work is meaningful for flood assessment and water resources management. However, the paper requires changes aimed at an improved presentation of the material before it can be accepted for publication.
- The introduction and discussion are not sufficient. Are there similar studies on other watersheds? The authors should compare their results with other studies to find the similarities and differences, and draw the generalized conclusions and recognize unique behavior of São Vicente’s watershed.
- It is not easy to quantify the roughness coefficient of the land cover. This manuscript modified the roughness coefficient to increase the drainage capacity. It is an idealized status. It is suggested to discuss the limitation.
- Line 398, delete a ‘in’.
- The legend of the figure 5 is confused. 970-1.117? or 1,117?
- There are two equation number “(1)”. The equation numbers of 6 and 10 are not in the correct order. Please reorder the equation numbers and check the corresponding context.
Author Response
Thank you very much for your constructive comments, allowing for a possible publication of the article, and enhancing the work developed in a peripheral and insular geographic location (Island of Madeira).
I will reply based on your 5 comments:
- Done - As I described in my introduction, I was able to validate the information of the Flood Risk Report prepared by DROTA, however, I proposed effective mitigation measures. Page 2, where I put the following information: Floods, heavy rainfall and the volume of water which is the source of, because of how much of each atmospheric origin seems, although determining the amount of water falling on the surface in terms of vegetation characteristics, influence the amount of water on the floor, underground infiltration of ground in terms of soil properties, plant and underground water leaking from the amount of the residual flow to the understanding of the causes of floods and geomorphology relationship is extremely important. There are also incorrect land uses, engineering structures on stream beds (urbanization, levees, embankments, dams, etc.) that assume an important role in the formation of floods. As this erroneous and often uncontrolled urban sprawl expands to rural areas, there is obviously the need to put structural and non-structural measures into action to prevent or at least mitigate floods impact [3-5]. A long time ago, the guideline would be to redirect the stream flow, changing the watercourse´s spatial disposition and subsequently its river mouth [5]. However, even though this principle is very effective in the upstream region, it worsens and increases flood risk downstream, therefore only benefiting half of the watercourse, population and assets whilst risking the other half. This concept does not solve the geomorphological and hydrological problem of the watershed which are often assembled with anthropic pressure. Consequently, there is the need for highly-impact measures to further mitigate floods impact and not only redirect the problem from one area to another [4,5,6,7]. In the Autonomous Region of Madeira, there is no study of this kind, my intention is in the near future to develop the same for the 27 watersheds in the region;
- Done - the change in the roughness coefficient is specific to the area at the mouth of the water line (stream), the part that is within the urban perimeter. We are not proposing a change across the entire catchment area. Page 2, I think that with the following description I demonstrate the innovation of the study: …structural measure was also chosen as it results in considerably reduced urban effects and can be complemented with small changes of the streambed and walls roughness coefficient, thus increasing the drainage capacity of the river mouth without affecting its cross section;
- Done - Implemented in red background in the article;
- Done - New figure, with scale modification, elimination of "dot";
- Done - Corrected the numbering of the equations throughout the article.
If it is necessary to process any adjustments or want any further clarification, I will be fully available. Thank you very much for your insight and analysis of this study.